# Childhood abuse and perinatal outcomes for mother and child: A systematic review of the literature

Robyn Brunton *

School of Psychology, Charles Sturt University, Bathurst Campus, Bathurst, NSW, Australia

* rbrunton@csu.edu.au

## Abstract

Childhood abuse can have long-term adverse outcomes in adulthood. These outcomes may pose a particular threat to the health and well-being of perinatal women; however, to date, this body of knowledge has not been systematically collated and synthesized. This systematic review examined the child abuse literature and a broad range of perinatal outcomes using a comprehensive search strategy. The aim of this review was to provide a clearer understanding of the distinct effect of different abuse types and areas where there may be gaps in our knowledge. Following PRISMA guidelines, EBSCO, PsychInfo, Scopus, Medline, CINAHL, PubMed, and Google Scholar databases and gray literature including preprints, dissertations and theses were searched for literature where childhood abuse was associated with any adverse perinatal outcome between 1969 and 2022. Exclusion criteria included adolescent samples, abuse examined as a composite variable, editorials, letters to the editor, qualitative studies, reviews, meta-analyses, or book chapters. Using an assessment tool, two reviewers extracted and assessed the methodological quality and risk of bias of each study. From an initial 12,384 articles, 95 studies were selected, and the outcomes were categorized as pregnancy, childbirth, postnatal for the mother, and perinatal for mother and child. The prevalence of childhood abuse ranged from 5–25% with wide variability (physical 2–78%, sexual 2–47%, and emotional/psychological 2–69%). Despite some consistent findings relating to psychological outcomes (i.e., depression and PTSD), most evidence was inconclusive, effect sizes were small, or the findings based on a limited number of studies. Inconsistencies in findings stem from small sample sizes and differing methodologies, and their diversity meant studies were not suitable for a meta-analysis. Research implication include the need for more rigorous methodology and research in countries where the prevalence of abuse may be high. Policy implications include the need for trauma-informed care with the Multi-level Determinants of Perinatal Wellbeing for Child Abuse Survivors model a useful framework. This review highlights the possible impacts of childhood abuse on perinatal women and their offspring and areas of further investigation. This review was registered with PROSPERO in 2021 and funded by an internal grant from Charles Sturt University.

**Data Availability Statement:** All relevant data are within the manuscript and its Supporting Information files.

**Funding:** This review was funded by an internal grant from Charles Sturt University. The funders

had no role in study design, data collection and analysis, decision to publish, or preparation of the manuscript.

**Competing interests:** The authors have declared that no competing interests exist.

## Introduction

Worldwide, millions of children suffer physical (CPA), sexual (CSA), psychological (CPY), or emotional abuse (CEA) that potentially have long-term deleterious outcomes [1]. While it is well established that the sequela of childhood abuse can be poor [1, provides a review], an important consideration is the impact of abuse on perinatal women. Certain adverse outcomes consistently linked to abuse could be exacerbated by pregnancy or impact the mother's well-being and behaviors. For example, if the risk of Post-Traumatic Stress Disorder (PTSD) and substance use is high for abuse survivors in the general population, it is important to know if this risk extends to pregnancy and postpartum. Moreover, particular experiences of pregnancy and postpartum can be exacerbated by child abuse, such as childbirth and breastfeeding [2,3], potentially impacting mother, child, and prenatal care.

Another consideration is that the body of knowledge regarding child abuse and perinatal outcomes needs to be collated to ensure that findings extrapolated globally are generalizable. Child abuse can exacerbate anxieties experienced during pregnancy [4], but these anxieties have wide cross-cultural variability [5]. Moreover, conceptions of child abuse can vary cross-culturally, with what constitutes physical abuse in one country deemed acceptable parenting behavior in another country [6]. Therefore, without a clear understanding of the body of knowledge, assumptions cannot be made that all findings apply to all pregnant women.

Few systematic reviews have synthesized research for all types of child abuse and a broad range of perinatal outcomes. To our knowledge, only one similar review examined CSA and pregnancy outcomes [7]. Other reviews have focused on child maltreatment more broadly [8] or specific outcomes such as mood disorders [9]. However, the distinct characteristics of different abuse types in the context of the unique developmental life stage of pregnancy, warrant separate examination. For example, the experience of CSA differs from other abuses; therefore, an individual with this history may avoid prenatal care for fear of intimate procedures, which may not be triggering for someone with a CPA history. Therefore, understanding the differential impact of child abuse on perinatal women is important in understanding their needs.

This review builds on previous reviews by expanded search criteria (i.e., all child abuse types and perinatal outcomes [i.e., conception to 12 months postpartum]) databases and gray literature. The aim is to provide a focused review of child abuse outcomes for perinatal women by collating and synthesizing the data to provide a comprehensive overview of the current knowledge. This targeted review will provide a clearer understanding of the distinct effect of different abuse types and areas where there may be gaps in our knowledge. Specifically, the following research questions were examined: Do pregnant and postpartum women with a history of child abuse have more adverse pregnancy, childbirth, and postpartum outcomes than other women? Do the offspring of pregnant women with a history of child abuse have more adverse outcomes than other children?

## Methods

The systematic review and protocol were registered with PROSPERO in 2021. PRISMA and other reporting guidelines guided the review [10,11]. The PICOT search framework mnemonic focused the question: Population = women who experience pregnancy and childbirth; Indicator = child abuse (includes any study that examines child abuse, noting that different studies may use different age limits to define abuse); Comparison = non-abused women; Outcomes = (1) pregnancy, (2) childbirth (3) child development including neonatal and infant, and (4) postpartum maternal outcomes; Time = perinatal (conception to 12 months postpartum).

Reviewer one [RB] and reviewer two ([DC], a research assistant) extracted the data. Studies were selected if they were original quantitative cross-sectional or longitudinal research

examining perinatal outcomes for women with a child abuse history. Child abuse could be a predictor or correlate of any adverse perinatal outcome in English-language peer-reviewed articles, dissertations, conference proceedings, or preprints between 1969 and 2022. Two searches were conducted, one in May 2021 and another in January 2023 to update the original 2021 search. However, the additional studies identified in 2023 were not subjected to two reviewers due to budget constraints.

Adolescent samples were excluded as they have unique psychosocial and biological challenges [12] however papers that included some adolescents in their sample [e.g., Ranchod et al., 2016] were included. Other exclusion criteria included abuse examined as part of a composite variable, editorials, letters to the editor, qualitative studies, reviews, meta-analyses, or book chapters. Databases searched included EBSCO (Academic Search Complete, Psychology and Behavioural Sciences and SocIndex), PsychInfo, Scopus (Elsevier), Medline, CINAHL, PubMed, and Google Scholar. In addition, gray literature searches included preprints (i.e., Pre-PubMed and OSFPreprints) and theses and dissertations (i.e., the ProQuest Dissertations and Theses Global database and EBSCO Open dissertations).

Reviewer one searched databases using the following search strategy to identify relevant articles: Child* AND (abuse OR maltreatment OR neglect) AND (Pregnan* OR antenatal OR Childbirth OR labour OR labor OR Neonat* OR Postpartum OR postnatal OR Perinatal). In addition, "history of violence against children" was also searched alongside the perinatal search terms, as it is used by many researchers globally [e.g., 13]. Consistent with 'back-in-time' pearling, reviewer one searched the reference list of each identified article using the 'find' function in the pdf viewer and the words child* and pregnan*. This search was conducted on articles after full-text screening. Data from the search results were downloaded into Endnote, which identified exact duplicates. All extracted records were imported into an Excel spreadsheet where reviewers independently used the inclusion/exclusion criteria to screen the records by title, abstract, and full text.

## Quality assessment

Once inter-rater agreement was achieved, the sample was quality assessed. An assessment tool developed from previous studies [(provided as S1 Appendix, 11,14] evaluated the methodological quality and risk of bias. Each study was rated on 1) selection procedures, 2) data collection methods, 3) study design, and 4) statistical analyses. Ratings were scored from 4 (*weak*) to 12 (*strong*). Consistency of ratings was confirmed by no substantial differences in the review of an initial ten articles. The remaining studies were individually assessed due to budget limitations. Consistent with Huang et al. [15], articles rated weak for all four domains were excluded from the review. In addition, the sufficiency of any sample $\leq 200$ was checked using G*Power [16] relative to the analyses conducted (i.e., < 5% chance of a Type 1 error). Those with insufficient power were excluded.

## Meta-analysis assessment

The homogeneity of the data was assessed using predefined criteria. Given the heterogeneity of the studies in terms of samples, design, predictors, and outcomes, no studies were deemed suitable [17]. S1 Table provides the criteria, full details of this assessment, and associated reasons for the unsuitability of the studies.

## Results

Fig 1 shows the PRISMA flow diagram. From 12,384 records identified, a sample of 555 studies was derived. For the initial search, the inter-rater agreement was 96.8% (disagreements

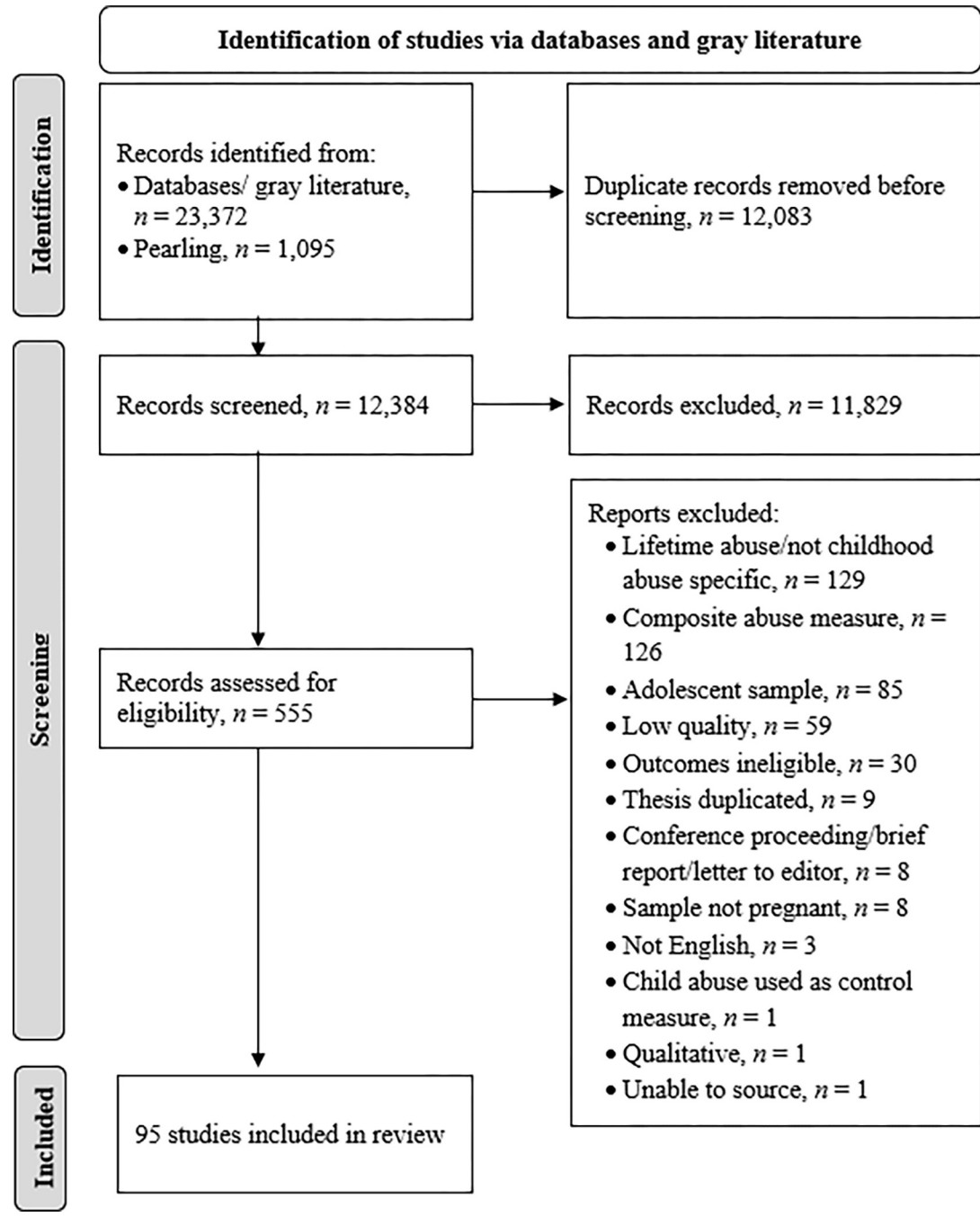

**Fig 1. PRISMA Flow chart of study identification and screening.**

resolved by discussion). For each iteration of pearling, the inter-rater agreement was 100%. Common reasons for exclusion were duplication of theses with published articles (theses retained for their detail), lifetime or current abuse examined, or using a summary/composite score. The final sample consisted of 95 articles.

The data extracted included the type of abuse, the child abuse measure, study details (e.g., longitudinal, country, participant age), perinatal outcomes, and findings. The outcomes were

categorized as pregnancy, childbirth, postnatal for the mother, and perinatal for mother and child. Records extracted represented studies from 25 countries with 643,241 participants (excluding duplicate samples); 59 studies were cross-sectional, 36 were longitudinal, and most were conducted in North America or other developed countries. Child abuse was commonly assessed using customized measures and the Childhood Trauma Questionnaire (CTQ) was the most used validated measure. The studies were of good quality ($M$ = 9.05 $SD$ = 1.42); most were rated low to medium bias (see S2 Table). S3 Table provides a detailed overview of the studies' characteristics (e.g., sample, abuse measure, prevalence, outcome measures, and quality rating).

## Prevalence

Consistent with Brunton and Dryer's [7] approach, prevalence estimates excluded samples < 200 (could lack external validity), with a high risk of abuse (may overestimate) and that utilized the same cohort as another included study. The prevalence of child abuse overall ranged from 5–25% with wide variability for the abuse types: CPA 2–78%, CSA 2–47%, and CEA/CPY 2–69%. The child abuse measure, conceptualization of abuse, and study location contributed to this variability.

## Findings from studies

For each subsection of the four main categories of findings, the main findings are summarized (in italics) when multiple studies are reported. Then, the more detailed findings are provided. S4 Table provides the main findings of each study.

## Pregnancy outcomes

### Unhealthy behaviors.

#### Disordered eating

*Two studies had findings related to pregnancy weight gain or control. The evidence indicates that all abuse types (i.e., CPA, CSA, and CEA) are related to weight or shape concerns or disordered eating. However, in adjusted analyses only CSA significantly predicts lifetime eating disorders and shape and weight concerns.*

Ranchod et al. [18] noted that at least two occurrences of CPA were independently associated with an increased risk of excessive gestational weight gain. Senior et al. [19] identified that CPA, CSA, and CEA independently predicted lifetime eating disorders and shape and weight concerns. However, only CSA remained significant after adjustment for other childhood factors (e.g., parental depression, support). In addition, CSA and CEA predicted antenatal eating disorder symptoms (e.g., self-induced vomiting), whereas CPA did not. However, these associations were not significant after adjusting for other childhood factors. In addition, all three abuse types were independent risk factors for shape and weight concerns during pregnancy (ORs = 1.56–1.78); however, after adjusting for childhood factors, only CSA remained significant for antennal shape and weight concerns.

#### Substance use

*Four studies examined smoking during pregnancy and predominantly identified CPA and CEA as risk factors for continued smoking. CSA, CSA with force, or CSA and depressive symtoms also predicted antenatal smoking. Fewer studies examined alcohol or drug use during pregnancy, and these indicate that CSA is a risk factor for antenatal drug use, and a history of any CPA or CSA increases the likelihood cocaine use (cf. no abuse).*

Blalock et al. [20] identified that smoking within five minutes of waking was more than twice as likely for pregnant women who experienced CPA and CEA (*cf.* women who reported

low/no trauma). Depression did not mediate these relationships. Cammack [21] identified that non-parental CSA motivated by physical force was associated with continued pregnancy smoking among women with a smoking history. Individuals with depressive symptoms and a history of caregiver CPA or non-caregiver CSA had the same risk level for continued smoking; however, those with 0–1 depressive symptoms and a CPA history were less likely to continue smoking. Nerum et al. [22] noted that, pregnant women subjected to CSA were more likely to smoke (*cf*. adult rape or no child abuse). Racine et al. [23] found that childhood family violence (i.e., CPA, CEA, and exposure to domestic violence) increased the odds of antenatal smoking, binge drinking, and drug use. A history of CSA was also a risk factor for antenatal drug use. Jantzen et al. [24], was the only study to examine illicit drugs and showed that women with a history of any CPA or CSA were more likely to use cocaine during their lifetime or perinatally (*cf*. no abuse).

### Maternal wellbeing.

#### Psychosocial risks

*Psychosocial risks were examined in six studies. Findings suggest that the individual abuse types alone or experienced jointly increased the risk of psychosocial difficulties (e.g., single parenting, being unsupported) relative to no abuse. However, the evidence relating to unplanned pregnancy is mixed with one study identifying child abuse as increasing the risk of unplanned pregnancy yet two studies failed to identify any dose-response relationship related to severity of abuse and unplanned pregnancy.*

Racine et al. [25] noted that women who experienced both CPA and CEA were more likelier to enter pregnancy with health risks and psychosocial difficulties than women who had not suffered abuse. However, no single abuse type significantly increased pre-pregnancy or reproductive health risks. Madigan et al.'s [29] study of 501 Canadian women identified that CPA or CSA increased the psychosocial risks (e.g., single parent) of pregnant women. One study [26] identified that women with a CSA history were less likely to participate in childbirth classes and, if they did participate, were unlikely to have a partner with them (*cf*. no abuse). These women also felt less prepared for labor and were less likely to have a trusted person with them during labor or feel they could participate in medical decisions. Similarly, Nerum et al. [22] found thatwomen with a CSA history were more likely to be unemployed or unsupported during pregnancy (*cf*. adult rape or no abuse).

Three studies examined unintended pregnancy. Dietz et al. [27] identified that less frequent abuse attenuated the risk of unintended pregnancy for CPA and CPY with multiple instances of abuse increasing the risk by 50%. Similarly, Drevin [28] noted that all abuse types, either individually or experienced jointly, and more frequent abuse was associated with an increased risk of unplanned pregnancy. This risk remained even when controlling for other abuses. Finally, Lukasse et al. [29] identified that CPA, CSA, and child abuse (any CPA, CSA, or CEA) predicted a high incidence of unintended pregnancy but no dose-response effect for severity was identified.

#### Biomedical risks

*Five studies examined five different biomedical risks. CSA was identified as a risk for persistent bacterial vaginosis and increased inflammation in pregnancy. Child abuse history predicted C-reactive protein and increased the risk of gestation diabetes mellitus (GDM). CPA increased the odds of thyroid dysfunction in postpartum women.*

Cammack et al. [30], in a sample of 312 pregnant US women, compared those who suffered CPA, CSA, or CEA to non-abused women and identified only CSA as a risk for persistent bacterial vaginosis (aOR = 3.07). Moreover, when stratified by race (a known risk factor for various adverse perinatal outcomes), the magnitude of this association was stronger for non-black people than black people. Finy and Christian's [31] study identified child abuse history as a

predictor of C-reactive protein. Pre-pregnancy body mass index (BMI) statistically explained this relationship.

Mason et al. [32] examined GDM with a large sample (*N* = 45,500). In adjusted analyses, different severities, and combinations of exposure of CPA and CSA increased the risk of GDM (*cf*. no abuse). Adjustment for pre-pregnancy BMI attenuated these associations slightly. Also, adjustment for not being overweight at 18 years also attenuated these associations and only severe CPA and CSA remaining significant predictors of GDM. CPA was more strongly associated with GDM if it occurred in adolescence than in childhood. Conversely, CSA was more strongly associated with GDM if it occurred in childhood than adolescence. Severe CPA and being overweight at 18 years increased the risk of GDM in a smaller subsample but no similar interaction for CSA was found.

Madigan et al.'s [33] study identified that CPA or CSA increased the biomedical (e.g., gestational diabetes) risks of pregnant women. Plaza et al. [34] found that only CPA increased the odds of thyroid dysfunction in postpartum women (*cf*. no abuse). Finally, Bublitz et al. [35] showed that a CSA history was linked to a greater change in the Neutrophil-lymphocyte ratio (NLR, a measure of systematic inflammation) over pregnancy.

### Cortisol

*Four studies examined cortisol with the evidence indicating that CSA is associated with elevated cortisol awakening response (CAR) in later pregnancy (cf. other or no abuse) and that this association may be moderated by family functioning and abuse severity. Also, experiencing joint CSA and CPA is linked to increased hair cortisol levels however this association was only evident among black women.*

CSA was associated with elevated CAR at 30 and 60 minutes after waking after controlling for proximal stressors in a small US study [36]. Bublitz and Stroud [37] found that 30 women with CSA histories had higher CAR at 35 weeks gestation than women reporting other or no abuse. In a second study of a similar size drawn from the same cohort, Bublitz et al. [38] identified that among participants with more experiences of CSA, those with poorer perceived family functioning had increased CAR at 35 weeks but not at 25 or 29 weeks compared to women with fewer CSA experiences and better-perceived family functioning. Schreier et al. [39] noted that women who experienced joint CPA and CSA had higher hair cortisol levels (*cf*. no abuse). When stratified by race/ethnicity, the association between child abuse and hair cortisol levels was only evident for black women.

### General health

*Seven studies examined aspects of general health or pregnancy-related health. Findings show that CPA or CSA are associated with poor past year health, hospitalization during pregnancy and more pregnancy complications. Also, CPA, CSA or CEA may increase the risk of current (e.g. migraine) or pregnancy-related health complaints or complications (cf. no abuse).*

In a study of postpartum women, Ansara et al. [40] noted that CPA or CSA did not predict bad headaches (cf. no abuse), however, CSA independently predicted backache, but this was no longer significant in adjusted analyses (e.g., depression). Perineal pain, hemorrhoids, fatigue, or bowel problems were not associated with CPA or CSA. Barrios et al. [41] identified that a history of child abuse increased the odds of reporting poor past year health or during the current pregnancy (*cf*. no abuse history). Similarly, women who reported CPA or CSA were more likely to have poor past year health and over twice as likely to have a poor past year and pregnancy health even after adjustment for current intimate partner violence (IPV).

Drevin [28], in a large cohort study, identified that women with self-reported CPA reported sacral and pelvic pain more often than those with no CPA. Lukasse et al. [42] examined 55,776 Norwegian women and found that women with CPA, CSA, or CEA were likelier to report seven or more common pregnancy complaints (e.g., heartburn, backache, *cf*. no abuse). In

addition, joint exposure to different abuses increased this likelihood. Sociodemographic characteristics and other risk factors did not explain this graded association. Finally, Littleton [43] found that those with a CSA history were more likely than nonvictims to report high levels of past month somatic complaint severity (e.g., headaches, indigestion). Depressive symptoms mediated this relationship (medium effect size). Controlling for sociodemographic variables resulted in similar findings.

Two studies [44,45] found that women exposed to CSA were more likely to be hospitalized during pregnancy (*cf.* no abuse). One study identified a link between CSA and reduced antenatal consultations after controlling for demographic and other key factors (e.g., dissociation, adverse childhood experiences). Women with a CSA history also presented more often with complications such as cervical insufficiency than other women. In adjusted analyses, women with a CSA or CPA history were likelier to have pregnancy complications.

Gelaye et al. [46] identified that any child abuse, CPA, or joint exposure to CPA and CSA increased the likelihood of migraine. In addition, a higher frequency of child abuse increased this risk. Adjusting for lifetime IPV attenuated these associations slightly. Yampolsky et al. [47] found that CSA survivors were almost twice as likely to have gynecological problems (*cf.* other trauma) after adjusting for depression and post-traumatic stress. However, a history of CSA or other trauma was not statistically significant in contributing to high-risk pregnancy.

### *Sleep*

*One study examined sleep and found that child abuse was related to stress-related sleep disturbance. Child abuse or CPA increased the odds of poor pregnancy sleep quality with a dose-response relationship relative to the frequency or joint exposure.*

Gelaye et al. [48] noted that women who experienced child abuse, multiple occurrences of abuse or jointly experienced CPA and CSA were more likely to have stress-related sleep disturbance than non-abused women. These associations remained after adjustment for age, ethnicity, and lifetime IPV. Experiencing CPA or CSA alone was not related to increased odds of stress-related sleep disturbance. When examining sleep quality, women reporting child abuse or CPA were around twice as likely to have poor sleep quality during pregnancy than other women. This risk increased relative to the frequency of abuse or if there was joint exposure to CPA and CSA. In separate analyses, antepartum depression and IPV mediated the relationship between child abuse and stress-related disturbance and sleep quality.

### *Memory*

Zhang et al. [49] examined memory impairment in a community sample. CPA, CSA, and CEA all predicted greater prospective and retrospective memory impairment. In adjusted analyzes, only CEA was independently related to both types of impairment. Women reporting both CEA and CPA had a higher prospective impairment, and those reporting both CEA and CSA or CEA and CPA had higher retrospective memory impairment. Women reporting all three types of child abuse also had higher prospective and retrospective memory impairment.

### *Miscarriage, stillbirth, and termination*

Atkinson [50] failed to find a relationship between CPA and stillbirth, miscarriage, or termination. Similarly, Freedman et al. [51] found no association between CPA, CSA, or CEA and stillbirth. However, in a smaller study [22] of women referred to mental health services, a history of CSA increased the likelihood of a previous termination or miscarriage (*cf.* adult rape or no abuse).

### Psychological and mental health disorders.

### *Antenatal depression*

*Nine studies examined antenatal depression. Findings strongly indicate that child abuse, CPA, CSA or a history of both CPA and CSA are risk factors for depression in pregnancy (cf. no abuse). In addition, race and a positive maternal relationship may influence the severity of depression. Only one study did not find significant results for CSA and depression.*

Barrios et al. [41] identified child abuse, CPA, or a history of both CPA and CSA as more than doubling the risk of depression in early pregnancy (*cf.* no abuse). This risk remained after adjusting for current IPV. CSA alone was not a significant risk factor. Likewise, Corona et al. [52] found a two-fold risk of depression in a sample of 382 low-income Hispanic/Latina women who reported child abuse. However, when stratified by country of birth, foreign-born Hispanic/Latina women had a lower depression risk (aOR = 1.96) than US-born women (aOR = 2.40).

Rich-Edwards et al. [53] also identified that a history of CPA or CSA increased the risk for antenatal depression in mid-pregnancy after adjusting for age and race. In addition, Yampolsky et al. [47] found that CSA survivors had a higher risk of depressive symptoms than others. Similarly, Chung et al. [54] noted that CPA and CSA increased the risk of depression (*cf.* no/other abuse). Of note, for CSA survivors without a positive maternal relationship, their risk of depressive symptoms more than tripled compared to women with no CSA and a positive maternal relationship.

Benedict et al. [55] found that contact or non-contact CSA was unrelated to antenatal depression symptoms but when they examined severe depressive symptoms, women who experienced CSA had a two-fold risk. Samia et al.'s [56] Kenyan study demonstrated that contact CSA predicted antenatal depression, consistent from early to mid-pregnancy. Galbally et al. [57] identified moderate-to-severe forms of child abuse increased the risk of an early pregnancy depression diagnosis but when examined separately, CSA posed over a four-fold risk. Only one study did not find a relationship between CSA and depression [58].

### Anxiety, worries and common mental disorders (CMDs)

Four studies had findings related to pregnancy-related anxiety, worries, and CMDs. Brunton et al. [4] demonstrated that CPA, CSA, and CPY all independently predicted pregnancy-related anxiety mediated by resilience and social support. Eide et al. [59] noted that women who reported CPA, CSA, or joint exposure were more likely to have strong worries about the baby's health than other women. Samia et al. [56] had no significant findings concerning CSA, pregnancy-related anxiety, and stress. Lydsdottir et al. [60] showed that CSA nearly tripled the odds of CMDs in pregnancy (i.e., depression or anxiety) after controlling for other traumas (e.g., poverty, being bullied). For CPA, however, the risk of CMD diagnosis was over five-fold.

### Suicide ideation

*Two studies examined suicide ideation finding that pregnant women who experienced any child abuse, CPA, CSA, or CEA had an increased risk of suicide ideation. The severity and frequency of abuse increased the risk exponentially. For women with a history of child abuse, depression or IPV potentially moderates this risk.*

Zhang et al. [61] identified that pregnant women who experienced any child abuse had over a three-fold risk of suicide ideation; controlling for depression slightly attenuated this. When examined independently, child abuse was associated with an increased risk of suicide ideation during pregnancy ranging from around a three-fold risk for CPA and CSA to close to five-fold for CEA. The severity and frequency of abuse increased the risk exponentially. Of note, women with a history of child abuse and depression had the greatest risk of suicide ideation (aOR = 17.78) than women without abuse or depression or women with only depression and no abuse history. Likewise, Zhong et al. [62] noted that any CPA or CSA nearly quadrupled the risk of suicide ideation in pregnancy compared to other women. This risk was attenuated by controlling for IPV and depression. Experiencing only CPA or only CSA increased the risk more than two-fold; however, joint exposure to CPA and CSA or multiple instances of abuse increased the risk more than four-fold, even after controlling for IPV and depression. The lack of a depression diagnosis did not reduce the risk of suicide ideation for those with a child abuse history.

*Post Traumatic Stress Disorder (PTSD)*

*Five studies examined PTSD during pregnancy. A history of CSA, CPA, CEA or joint CPA and CSA increased the risk for PTSD or related symptomology and this risk increased exponentially for those who experienced child abuse and IPV. Only one study failed to find any significant findings related to the individual abuse types and PTSD.*

Yampolsky et al. [47] found that a history of CSA doubled the risk for high post-traumatic stress symptoms. Sanchez et al. [63], in models adjusted for age, ethnicity, and lifetime IPV, showed that women who experienced CPA or CSA had an increased risk of PTSD (*cf.* other women). This risk increased eightfold when CPA and CSA were experienced together. Also, women who experienced child abuse were more than five times more likely to have PTSD and experiencing child abuse and IPV increased the odds of PTSD to more than twenty-fold.

Seng et al. [64] examined primiparous women using cluster analysis. CPA was associated with an increased likelihood of affect/relational dysregulation, and CSA doubled the odds of comorbid PTSD. In addition, the odds of being in the healthy cluster for both abuse types were low (aOR < 0.62). Huth-Bocks' [65] smaller study examined women in late pregnancy. After controlling for age and income, there were no significant findings related to the individual abuse types and PTSD. Finally, Diestel et al. [66] examined child abuse and PTSD symptoms with women from an IPV and epigenetic risk study. All types of child abuse were associated with PTSD symptomology, with the strongest findings for CEA. This abuse type was strongly associated with social isolation with weaker associations for intrusive thoughts and negative emotionality.

## Childbirth outcomes

**Birth complications.** *Four studies examinend bith complications with the findings indicating that women with a CPA or CSA history had increased risk of perinatal and obstetrical complications and more likely to have a difficult birth, medical interventions during birth, or their child transferred to intensive care. CPA and CSA were also independent risk factors for complicated delivery.*

Shamblaw et al. [67] found that women with a CSA or CEA history (*cf.* no abuse) had around a two-fold increased risk of perinatal and obstetrical complications. However, after adjustment for lifetime psychiatric disorders, only CSA was significant. Heimstad et al. [68] noted that CPA and CSA were independent risk factors for complicated delivery (i.e., operative vaginal delivery or caesarean); both posed around a 2.5 increase in risk than women with no abuse history. Leeners et al. [26] showed that pregnant women with a history of CSA were more likely to have a difficult birth and less likely to deliver spontaneously. Finally, Nerum et al. [22] found that relative to women who experienced adult rape or no abuse, women with a CSA history were more likely to have obstetric risks, induced or augmented labor, epidural analgesia, or their child transferred to intensive care.

**Fear of childbirth.** *Three studies had findings related to a fear of childbirth. Consistently, child abuse survivors have been shown to have more childbirth fear than other women regardless of parity. A previous negative birth experience increases the risk exponentially. CSA history increases the likelihood of intense fear of delivery, negative birth perceptions, and a greater fear of delivery.*

Lukasse et al. [3] found that regardless of parity, child abuse survivors feared childbirth more than other women. However, primiparas with a CPA, CSA, or CEA history were more likely to fear childbirth severely (*cf.* no abuse). The severity of abuse also increased the odds of the severity of fear of childbirth. Experiencing simultaneous abuse increased the odds, and a collective history of all three abuse types posed the greatest risk for severe fear of childbirth

(aOR = 5.30). For multiparas, only non-contact, and mild CSA (e.g., humiliation) increased the odds of severe fear of childbirth. CPA, severe CPA, CEA, and any child abuse likewise were risk factors. Also, when adjusted (e.g., demographics, adult abuse, previous negative birth experience), the association between child abuse and severe fear of childbirth attenuated slightly for primiparas but was not significant for multiparas. However, multiparas who experienced any child abuse and had a negative birth experience were nearly six times more likely to have a severe fear of childbirth than women with a history of child abuse and a positive birth experience. However, no child abuse and a negative birth experience posed a much higher risk (nearly nine times) for severe fear of childbirth.

Lukasse et al. [69] examined multiparas (second pregnancy). Child abuse increased the risk for fear of childbirth after adjusting for sociodemographics, adult abuse, previous birth experience, and perception of first pregnancy. However, only CEA was a significant risk factor for fear of childbirth. Finally, Leeners et al. [26] demonstrated that women with a CSA history were more likely to have an intense fear of delivery than other women, twice as likely to have a negative birth perception, and nearly three times more likely to fear delivery.

**Low birthweight.**  Hyle's [70] study of low-income women found that a history of CSA (broadly defined), predicted birth weight, whereas CPA did not (*cf.* no abuse). Additionally, specific abuse characteristics (e.g., onset age) predicted birth weight. Leeners et al. [45] identified that women with a CSA history were more likely to have a low birthweight baby (< 2800 grams) than non-abused women. In contrast, Benedict et al. [55] did not find differences in birth weight or baby characteristics for women exposed to CSA compared to those who were not.

**Mode of delivery.**  *Five studies examined the mode of delivery. The finidngs show that regardless of parity, a history of abuse was not associated with operative delivery, but CPA was associated with a significant increase in emergency cesareans for multiparas. Moreover, any child abuse or CSA was linked to increased cesarean during labor and women who suffered CPA or CSA were more likely to desire a cesarean before their second pregnancy birth.*

Schei et al. [71] utilized data from a multi-country study, finding that regardless of parity, a history of abuse was not associated with any operative delivery. However, for multiparas, a history of CPA was associated with a significant increase in emergency cesareans, and this association attenuated after controlling for previous cesarean. There was no association between CPA and non-obstetrically indicated cesarean. In contrast, Nerum et al. [22] found that women with a CSA history were no more likely to have an operative vaginal delivery or cesarean than women with a history of adult rape or no abuse. Benedict et al. [55] failed to find significant differences between CSA and no CSA for the type of delivery (e.g., caesarean, induction). Lukasse et al. [72] identified that more women with an abuse history had an increased chance of caesarean during labor than other women. However, none of the distinct categories of child abuse was significantly associated with a cesarean before childbirth; however, any child abuse and mild CEA were associated with increased cesarean during labor compared with women with no child abuse. After adjustment, only any child abuse remained a risk factor for cesarean during labor. Using a smaller subsample from the same cohort [69], women who suffered any CPA or CSA were more likely to desire a cesarean before their second pregnancy birth than women with no abuse.

**Preterm birth (PTB).**  *Six studies have findings related to PTB however most have failed to identify child abuse or specific types (e.g., CPA or CSA) as risk factors for PTB. In contrast one study found that a CSA history was associated with an increased risk of PTB (cf. no abuse), and this risk was heightened for women from a race other than Hispanic, black, or white. Only one study identified that the frequency of child abuse or exposure to child abuse and IPV, doubled the risk of placental abruption.*

Cammack [21] found that women with CSA perpetrated by a non-parent and motivated by non-physical threats had nearly double the risk of very PTB (before 34 weeks) than women with no child abuse. Further, women from a race other than Hispanic, black, or white were nearly eight times more likely to have a very PTB. Moreover, the risk for 'other race' and birth before 37 weeks nearly quadrupled when they had experienced the same type of CSA (*cf.* no abuse). Leeners et al. [44,45] conducted two studies with the same sample. Women with a CSA history were more likely to have a PTB or premature contractions.

In contrast, studies have more consistently not identified child abuse or specific types (e.g., CPA or CSA) as risk factors for PTB [70,73,74]. However, Hyle [70] showed specific abuse characteristics (e.g., more than three incidents) predictive of lower gestational age in a sample of low-income women. Specifically, there was an associated decrease in gestational age for each additional abuse incident. Only one study found that any child abuse and CSA increased the chances (1.5 times) of late PTB at 35–36 weeks [74].

Mitro et al. [75] conducted the only study on placental abruption. Child abuse did not increase the odds of placental abruption in adjusted models for demographics and parity. However, experiencing three or more instances of child abuse increased the odds of placental abruption by just over 1.5 times. In addition, joint exposure to severe child abuse and IPV doubled this risk compared to rare or no abuse. When examined among women delivering preterm, experiencing three or more child abuse events almost doubled the risk for placental abruption. Similarly, among women delivering full-term babies, experiencing joint and severe exposure to childhood abuse and IPV was associated with close to a three-fold risk of placental abruption.

## Postnatal outcomes for mothers

**Mother-child relationship.** *Five studies examined attachment/bonding with the evidence suggesting that CEA and CSA are associated with less maternal-fetal attachment or bonding disturbances. However, postpartum no specific abuse types predict mother-infant bonding impairment.*

Two studies identified CEA as a predictor of lower maternal-fetal attachment [76] and mother-infant bonding disturbances [77]. Also, Nieto et al. [78] found that CSA (*cf.* no CSA) nearly tripled the odds of lower maternal attachment. In contrast, Lehnig et al. [79] did not identify specific abuse types that predicted mother-infant bonding impairment at two months postpartum. However, their result may reflect their sample's low prevalence of abuse. Choi's [80] study of South African mothers did not identify any abuse type significantly affecting maternal-infant bonding in analyses controlling for other abuse types.

**Parental practice.**

*Self-efficacy and parenting morale*

Two studies examined parenting outcomes. Kunseler et al. [81] identified that postpartum women who reported child abuse did not differ in parenting self-efficacy or changes in their efficacy after positive feedback (than other women). However, for those who reported child abuse, their parenting self-efficacy decreased after exposure to a difficult-to-soothe infant (*cf.* non-abused women). Malta et al. [82] noted that women with a child abuse history were nearly twice as likely to have low parenting morale.

*Child abuse harm or potential*

*Three studies examined child abuse potential or harm. Findings indicate that a history of CPA, CSA or CEA are associated with an increased likelihood of harm to the infant/toddler and this association may be mediated by substance abuse or depression. Moreover, for first-time mothers those with a history of CPA or CEA, they may be less responsive or have reduced empathic awareness in parenting, have harsher punishment attitudes and be more accepting of child abuse/neglect.*

Appleyard et al. [83] demonstrated that a maternal history of CPA or CSA increased the likelihood that the mother would victimize their infant/toddler. Maternal substance abuse mediated this relationship. Bert et al. [84] noted that CEA predicted less responsivity/empathic awareness in parenting, harsher punishment attitudes, more acceptance of child abuse/neglect, and greater child abuse potential in first-time mothers. CPA predicted the same outcomes except for child abuse potential, which was only significant for mothers older than 21 with some college education. Choi [80] identified CEA as indirectly influencing child harm exposure through postpartum depression (PPD). CPA or CSA were not significant contributors.

**Postpartum psychological outcomes.**

*Postpartum depression (PPD)*

*Ten studies had findings relating to PPD. Women who experienced child abuse may be more likely to have PPD with the risk increasing exponentially for multiple types of abuse or experiencing child abuse and pregnancy IPV. Three studies found that CEA increased the risk of PPD. In contrast, one study failed to find any association between CPA, CSA, and CEA and PPD but may have been limited by its sample.*

Mahenge et al. [85] identified that women who experienced child abuse were between 2.5 and over five times more likely to have PPD. The risk nearly quadrupled for those with a joint history of CSA and CPY and increased to more than nine times if they experienced child abuse and IPV during pregnancy. Bahadur et al. [86] noted that CEA increased the risk of PPD nearly seven-fold after adjusting for sociodemographic factors, whereas CPA and CSA were not significant. However, a larger Australian study [87] showed a similar but lower risk for CEA (aOR = 1.4). Nagl et al. [88] found that women with severe CEA were nearly 20 times more likely to have PPD. Severe CPA or CSA, slight and moderate CPA, or CEA were lesser risk factors. Most of these associations remained even after controlling for other abuses. One Ethiopian study [89] identified that a history of CSA was a stronger predictor of PPD than other sociodemographic variables. In studies that examined all abuse types, CPA, CSA, and CEA independently predicted PPD; however, only CPA remained significant when controlling for other abuses [34]. CEA and CSA were also risk factors for early-onset PPD, and CEA, CPA, and CSA were all risk factors for late-onset. The risk remained for CSA and early-onset PPD and CEA and late-onset PPD after adjusting for sociodemographic factors and other abuse types [90]. In contrast, a South African study showed that only CSA and CEA predicted PPD [80]. Finally, Cohen et al. [91] demonstrated that CPA, CSA, and CEA was not predictive of PPD however insufficient participants with abuse experiences may have impacted this result.

*Postnatal anxiety, stress and PTSD*

*Three studies examined postnatal anxiety or stress finding that child abuse was not linked to postnatal anxiety or stress. However, when examined by subtype CPA, CSA and CEA were independent risk factors for postnatal anxiety. One study failed to find any association between CSA and PTSD symptoms hypothesising an indirect effect.*

Malta [82] did not identify any association between child abuse and postnatal anxiety or stress. In contrast, Nagl et al. [88] identified CPA and CEA as independent risk factors for postpartum anxiety with severe CSA associated with a nearly seven-fold risk. These associations were lower after controlling for other abuses. Mayhew and Thomas [92] noted that prenatal anxiety was the strongest predictor for postnatal anxiety. CEA and distressing childbirth were also significant predictors in the same model.

Oliveira et al. [93] conducted the only study of postnatal PTSD symptoms noting that CSA was related to other traumatic events such as IPV and fear of childbirth but not directly related to PTSD symptoms suggesting an indirect effect.

**Sleep.** Swanson et al. [94] found that women with a CPA or a joint history of CPA and CSA had greater difficulty falling asleep. Of interest, the risk was greater for CPA alone

(aOR = 9.20) than for experiencing CPA and CSA jointly (aOR = 5.95). In addition, women with a history of CPA or those who experienced CPA and CSA simultaneously were more than three times more likely than other women to have difficulty staying asleep.

## Perinatal outcomes for mothers and newborns

**Offspring asthma and autism.** Only one study examined asthma but did not identify significant associations with maternal CPA or CSA [95]. Roberts et al. [96], the only study to examine autism risk, found that severe CSA doubled the risk for offspring autism, with severe joint exposure to CPA and CEA a lesser risk (*cf.* no abuse). In addition, experiencing multiple instances of child abuse tripled the risk of autism regardless of the child's biological sex. In adjusted models, gestational diabetes, previous abortion, and smoking mediated the relationship between child abuse and autism.

**Breastfeeding.** *Breastfeeding was examined by three studies with the findings indicating that women who experienced child abuse were more likely to stop breastfeeding earlier than other women. Also, a history of CSA increased the likelihood of mastitis or painful breastfeeding which may be a contributing factor to early cessation.*

Sorbo et al. [97], using a large sample, noted that women who experienced child abuse or CSA were more likely to stop breastfeeding before four months. The odds of breastfeeding cessation by four months were lower for CPA and/or CEA. Coles et al. [2] identified CSA survivors as less likely to breastfeed for longer than six months however, this result was no longer significant after adjustment for adult violence and other factors (e.g., smoking, partner status). Elfgen et al. [98] noted that a history of CSA nearly tripled the likelihood of mastitis or experience painful breastfeeding (OR = 5.77). However, there was no difference in breastfeeding prevalence between these women and those with no CSA history.

**Intimate Partner Violence (IPV).** *Three studies examined IPV identifying that CPA, CSA, and CEA increased the likelihood of IPV in the past-year with specific abuse types linked to specific IPV subtypes. Child abuse, CPA, and CSA were risk factors for any IPV with joint exposure (i.e. CPA and CSA) increasing the risk. However, when other types of violence are controlled, the results were non-significant suggesting a mediating effect. Only one study did not identify child abuse (CPA and CSA) predictive of IPV in a sample of HIV-positive women.*

Barnett's [99] longitudinal study (pregnancy and postpartum) demonstrated that CPA, CSA, and CEA increased the likelihood of high past-year IPV. When specific IPV subtypes were analyzed, a CPA history was a higher risk for moderate emotional and sexual IPV and high physical IPV (*cf.* women with low/no child abuse). CSA and CEA were associated with high emotional or physical IPV and high/moderate sexual IPV.

Barrios et al. [41] identified that child abuse, CPA, and CSA were risk factors for any IPV and physical lifetime IPV; joint CPA and CSA posed the highest risk. Child abuse also predicted lifetime sexual IPV, and a history of both CPA and CSA posed a five-fold risk of this IPV type. Child abuse and CSA predicted both physical and sexual lifetime IPV. A history of CPA and CSA increased this risk more than seven-fold. Comparable results were evident for IPV in the past year. CPA and CSA were associated with greater odds of revictimization of physical IPV. CSA was also related to sexual IPV revictimization. Experiencing both CPA and CSA increased the risk of physical and sexual lifetime IPV revictimization. However, the greatest risk of revictimization was for those who experienced CPA and CSA and combined physical and sexual IPV, aOR = 6.88.

Castro et al. [100] identified a higher prevalence of pregnancy IPV for those who experienced moderate to high CPA or CEA (*cf.* no/low abuse). Both abuse types more than doubled the risk of IPV; however, when other types of violence were controlled, the results were non-

significant. Only one study did not identify child abuse (CPA and CSA) predictive of IPV in a sample of HIV-positive women [101].

**Maternal psychological outcomes.**

*Depression and anxiety*

Seven studies examined depression across the perinatal period. The findings show that child abuse, CPA, CSA, and CEA all increased the risk of antenatal or postnatal distress or depression or high depressive symptoms. One longitudinal examination showed that women who experienced CSA had higher anxiety symptoms during pregnancy and postpartum than non-abused women.

Kang et al. [102] demonstrated that child abuse nearly doubled the risk of peripartum depression in a large Turkish study. Li et al. [103] found that women who experienced CSA or CEA were more than four times more likely to have antepartum depression than non-abused women. Also, women with a CEA history had higher depression scores in pregnancy which lessened over the perinatal period than women with no CEA, whose scores started lower but increased postpartum. In fact, the depression scores for women with a CEA history, while higher in pregnancy, were much lower than those with no CEA by four weeks postpartum. Also, no significant effects for any abuse or frequency of abuse was found for PPD scores, or between abuse types and chronic depression risk.

Akinbode's [104] study showed that CSA predicted PPD symptoms in pregnancy, six and 12 weeks postpartum. CPA predicted depression in pregnancy and 12 weeks postpartum but not six weeks postpartum. Experiencing both CPA and CSA also predicted antenatal depression, but there was no significant finding for PPD. In contrast, Robertson-Blackmore et al. [105] found that women who reported CSA were more likely to report antenatal depressive symptoms than other women but were not more likely to report PPD. CPA/parental neglect was not significant for antenatal depression or PPD.

Khanlari et al. [106] compared women with a maternal history of abuse with other women, noting that child abuse increased the risk for antenatal and postnatal distress and depression. The highest risk was for antenatal depression (aOR = 3.2). Similarly, Ogbo et al. [107] examined a large sample of culturally and linguistically diverse (CALD) Australian women. Those with a child abuse history had nearly double the risk for antenatal distress and were more than twice as likely to have antenatal or PPD than non-abused CALD women. Giallo et al. [108] noted that CPA and CSA independently increased the risk for subclinical and persistently high depressive symptoms for nulliparous women. However, CPA was the only significant risk factor for subclinical and persistently high depressive symptoms after controlling for sociodemographic factors.

Akinbode [104] conducted the only longitudinal examination of anxiety, noting that women who experienced CSA had higher anxiety symptoms during pregnancy, six and 12 weeks postpartum, than non-abused women. CPA and experiencing both CPA and CSA were not significant predictors.

*Post Traumatic Stress Disorder (PTSD)*

*Three studies examined PTSD. The evidence shows that women with a CSA history or those who experienced CPA and neglect have a higher risk of lifetime PTSD. Greater exposure to child abuse also predicts antenatal PTSD symptoms and a history of CSA predicts greater PTSD symptomology, postpartum.*

Robertson-Blackmore et al. [105] showed a higher risk of lifetime PTSD for women with a CSA history (aOR = 9.21) or those who experienced CPA and neglect (12.16) in models adjusted for demographics. Also, in adjusted analyses, Sumner et al. [109] found that higher exposure to child abuse predicted antenatal PTSD symptoms but not postpartum symptoms. Finally, Lev-Wiesel et al. [110] noted that a history of CSA predicted greater disassociation at

two months postpartum and more birth-related PTSD symptoms, avoidance, and arousal levels at seven months postpartum.

## Discussion

### Summary of findings

Despite some variability in the findings, predominantly studies confirm that a history of child abuse is a significant risk factor for poor pregnancy outcomes. CSA was the most examined abuse type consistently linked to pregnancy and childbirth outcomes. These outcomes included substance use (i.e., alcohol, smoking) which is of critical concern given the harmful teratogens introduced into the prenatal environment. However, none of the studies examined alcohol as an independent outcome; this area requires more attention. Poor mental health was the focus of several studies conducted in various countries, supporting the generalization of the findings. There were consistent findings that CPA and CSA, independently or jointly, represent a greater risk of antenatal depression and PTSD symptoms during pregnancy.

Studies examining biomedical outcomes confirm that child abuse increases the risk for certain conditions (e.g., enhanced inflammatory states) linked to poor maternal and neonatal health and perinatal complications. Also, CSA was associated with higher cortisol levels, whereas child abuse overall or different subtypes had less consistent findings. High cortisol levels may increase the HPA axis' sensitivity to maternal distress, shorten gestation and increase antenatal health problems such as preeclampsia or gestational diabetes [38]. For the child, increased prenatal cortisol is associated with poor developmental outcomes [e.g., restricted growth, 111]. However, the findings should be viewed cautiously, given the small samples used.

Several studies established a link between child abuse and aspects of poor general health but not all types of poor health were linked to abuse, suggesting that the type of health condition may play a pivotal role. For example, a history of CSA is linked to gynecological problems in areas consistent with the foci of that abuse type. In contrast, CPA is associated with greater pelvic or sacral pain; these are areas that could be targeted during physical abuse. Also, findings that abuse survivors can have more common health complaints are consistent with trauma leading to a vulnerability for poorer mental and physical health and, for some, unexplained complaints [112]. Related to health is sleep disturbance, with studies establishing that women with a history of child abuse are more likely to have poorer sleep quality. Despite sleep disruption normative during pregnancy, poor sleep can contribute to poor outcomes [113, provides a review]. Finally, findings regarding care indicate that women with a CSA history are more likely to be hospitalized during pregnancy or have reduced consultations. While these results may seem contradictory, it is conceivable that a lack of standard care may lead to more serious problems requiring hospitalization. However, caution is warranted in interpreting these findings due to the small effect sizes.

### Childbirth

All abuse types increase the risk of delivery complications, and these have largely been studied for CSA. Child abuse survivors also have a greater preference for a cesarean; however, while this preference exists, the chances that it occurs are low. Multiparous women with a CPA history are also more likely to have an emergency cesarean which may be related to previous abdominal trauma. The finding that controlling for any previous caesarean attenuated this risk supports this proposition.

PTB and low birthweight have attracted the most interest from North American researchers using obstetric data (than self-report), which adds certainty to the findings. Predominantly

CSA is associated with low birthweight more than other abuses. However, studies are limited by not considering the influence of other related factors identified in this review (e.g., self-care, eating disorders, and substance use) which could contribute to less nourishment for the fetus. Therefore, the mechanisms by which low birthweight occurs are unclear. Likewise, the association between CSA and PTB is inconsistent, suggesting there may be significant mediators or moderators of this relationship. However, again there are limited studies in this area.

## Postnatal outcomes for mothers

Bonding and attachment are critical developmental tasks of motherhood [114], yet findings are inconsistent and require cautious interpretation (from single studies with small effects or unrepresentative samples). Given that several factors have been identified that result in a woman psychologically distancing from her pregnancy and newborn [e.g., anxiety and depression, 115], an examination of other indirect influences is needed.

Child abuse is a risk factor for low parenting morale [82] and predicts less parenting self-efficacy after exposure to a difficult-to-soothe infant. This large effect size for self-efficacy in nulliparous women indicates that these mothers may lose belief in themselves more readily in stressful parenting situations [81]. Relatedly, CPA, CSA, and CEA predicted exposure or potential to harm their child. Of note, substance use and PPD mediated the relationship between CEA and child harm exposure between the ages of five and 12. The authors explained that this delayed onset of maltreatment of abused mothers' offspring is due to the persistence of depression beyond the postpartum period. These findings are particularly important given that vicarious learning may model later parenting behaviors [116], contributing to a generational cycle of violence.

Similarly, there are consistent links between CPA, CSA, and PPD. Less consistent are the findings for CEA, CPY and PPD. Only one study [91] did not find a relationship between CSA and PPD and CEA and PPD; however, their smaller sample and modified abuse measure may have limited findings. There were some consistent findings that CPA, CSA or CEA increase the risk of perinatal anxiety, with contradictory findings likely due to the purpose-created abuse measures used [e.g., 82].

Only one study examined CSA and postnatal PTSD symptoms, with the results trending toward significance ($p$ = .07), consistent with longitudinal studies across pregnancy and postpartum. Of interest, while identifying that PTSD symptoms have a downward trajectory from pregnancy to postpartum, Lev Weisel and colleagues' comprehensive study also showed that CSA predicts significant PTSD symptoms seven months postpartum. Prima facie, this downward trajectory does not seem consistent with theorizing that childbirth for abuse survivors is retraumatizing; however, symptoms relating to intrusion and arousal, which had a time interaction, may indicate a delayed response to childbirth. Alternatively, the thought of childbirth may be retraumatizing for some abuse survivors, and the effects of this may lessen if the experience was not traumatic.

## Perinatal outcomes for mothers and newborns

CSA posed the greatest risk for breastfeeding cessation, with CEA or CPA also identified as a risk for earlier cessation. One study identified that a history of CSA was related to painful breastfeeding, which may contribute to decisions to stop breastfeeding earlier, although this outcome was not examined. In addition, the link between child abuse, particularly CSA and breastfeeding, could be related to breastfeeding involving skin-to-skin contact, triggering memories of abuse. Given the importance of breastfeeding for at least six months to optimize an infant's growth, development, and health and the benefits for the mother [e.g., reduces cancer risks, 117], this is an understudied area that requires more attention.

The risk of revictimization of survivors of child abuse is high with consistent links between child abuse, its subtypes and lifetime, past year, or current IPV [41,99,100]. These findings are consistent with child abuse, a known risk factor for adult revictimization [118]. The one study that did not identify abuse as a risk of revictimization utilized an HIV-positive sample and a comparison of no or low abuse, possibly limiting their findings [101]. During pregnancy, this risk for IPV may increase, potentially endangering the expectant mother and the unborn baby. This risk is explained by experiential avoidance, which proposes that dissociation and PTSD symptoms influence revictimization [119].

Results indicate that CSA may be a more consistent predictor of depression across the perinatal period than CPA or CEA. One study identified an interaction between CEA and time with women who experienced emotional abuse; their depression scores decreased across the perinatal period compared to women who had not suffered this abuse, whose scores increased. There was no similar interaction for other abuse types. No studies examined CPY.

## Methodological issues and implications for research

The CTQ was the most common child abuse measure utilized by 24% of studies. This questionnaire has adequate psychometric data; however, the short-form version, used more frequently, needs further validity evidence [7]. Many studies relied on purpose-created measures or tools without documented reliability and validity to assess child abuse. This significant issue potentially impacting findings and cross-study comparisons.

Despite consistent findings across the studies, effect sizes were small or negligible in some cases, likely due to the samples used or the methodology employed. Some studies used small convenience samples that can amplify effects or are unrepresentative of the population. In contrast, several studies analyzed data as part of large-scale projects. Despite the common belief that larger sample sizes are superior to small samples, excessive sample sizes can exaggerate small or even non-significant effects [120]. To provide more certainty for future findings, researchers should determine the sample size *a priori* to ensure it is adequate for the planned analyses. Indeed, this review excluded 36 studies with insufficient sample sizes relative to the power to conduct their analyses. Also, it is important to acknowledge that some of the pathways between child abuse and the outcomes reported are complex. However, few studies identified in this review examined mediation/moderation or interact effects, focussing instead on the direct relationship however the influence of other variables needs to be considered when examining the sequalae of child abuse.

Eighteen studies examined only one type of abuse, yet child abuse is rarely isolated, with many victims subjected to multiple maltreatment [121]. In contrast, when examining multiple types of abuse, there was a failure to partition other abuse types statistically [e.g., 40], which could overestimate findings as it cannot be assumed that the effects noted are specific to the abuse examined. Moreover, the propensity to examine abuse as a combined variable limits our understanding of the impacts of certain abuses in the perinatal period. Relatedly, few studies included adequate control of other variables, such as socioeconomic status or family environment previously identified as risk factors for child abuse [122]. These variables could interact or account for some of the variance, and when not included, effect sizes may be overestimated. This highlights that the design of child abuse studies requires careful consideration to ensure that the findings reflect the population of interest.

The studies captured by this review were predominantly North American, with an over-focus on CSA. This focus is likely an artifact of historical understandings of abuse, with CSA considered part of battered child syndrome some 45 years ago [123]. However, child abuse is a worldwide phenomenon with wide variability in the prevalence of abuse types across different

countries. Generally, CPA is more prevalent in countries such as South America and Africa, CSA in North America and Australia, and CEA and CPY in South America and Australia [124]. While this disparity may relate to differing conceptualisations of abuse (e.g., [punitive parenting practices may be acceptable in some cultures and not others), or differing methodologies for assessing abuse [125] they also highlight the disparity between the countries where child abuse is predominantly studied and the absence of studies in countries with a higher abuse prevalence (e.g., Africa and South America).

Also, CPY and CEA are often treated as synonymous, likely stemming from cognition and emotion considered interdependent [126]; examining them independently or statistically controlling them may clarify inconclusive findings. Indeed it can be argued that CPY and CEA are inherent in all abuse types and may even be more prevalent than CPA or CSA [(see 42, for example].

## Implications for policy

Child abuse outcomes related to pregnancy (e.g., poor physical and mental health and unhealthy behaviors), highlight the importance of early prenatal screening. Australian, UK, and US guidelines recommend screening for abuse or trauma, but not all specifically name child abuse [127]. However, given the identified barriers and challenges of child abuse screening (e.g., the need for sensitive inquiry), many researchers and clinicians advocate trauma-informed care for perinatal women [127,128].

The Multi-level Determinants of Perinatal Wellbeing for Child Abuse Survivors model [129] provides a framework for this care using these principles that engender empowerment, safety, and trust. All women can have a supportive environment through universal screening, leading to referral and intervention. Importantly this model includes the needs of both the survivor and clinician. For example, a child abuse survivor requires a safe and appropriate environment to help facilitate disclosure, whereas a clinician needs training, time, and setting considerations. Therefore, policies around screening need to be based on trauma-informed principles that address the needs of the clinician, survivor, and environment [127–129].

While identifying women with a history of child abuse is important in affording more sensitive antenatal care, for many women, postpartum care is focused on physical recovery and less on psychosocial or psychological factors [130]. However, child abuse is also a risk factor for postpartum outcomes such as poor mental health, child abuse potential, and IPV. However, there is limited opportunity to identify child abuse and intervention without continuity of care extending to the postpartum period.

## Strengths and limitations of the review

The strength of this review was its comprehensive examination of a wide range of databases and grey literature. In addition, the double screening of search results and rigorous quality assessment contributed to the accuracy of the data. This search strategy enabled a wide range of evidence to be synthesized, thus contributing to the body of knowledge. Notwithstanding this, the review was potentially limited by excluding non-English language studies limiting the generalizability of the findings. Also, the quality assessment was mostly conducted by one rater (due to budgetary constraints) which is a limitation that should be noted. Furthermore, as detailed above, many studies reviewed had methodological limitations and used retrospective and self-report data for child abuse/maltreatment, limiting the ability to draw causal conclusions.

## Conclusion

This comprehensive review examined outcomes related to child abuse for perinatal women. Notwithstanding this, this review demonstrated that maternal child abuse has implications for

perinatal women. Perinatal women may be at higher risk for poorer mental health, difficulties with childbirth, and the experience of prenatal care and motherhood. For the child, their mother's previous history may have implications for their healthy growth and development and intergenerational implications through the transmission of violence. Identifying women at risk is the first step in positive interventions, which may best be achieved through care based on trauma-informed principles.

## Supporting information

**S1 Appendix. Data extraction sheet for quality assessment and risk of bias.**
(DOCX)

**S1 Table. Assessment of the studies for inclusion in the meta-analysis.**
(DOCX)

**S2 Table. Overview of the included studies.**
(DOCX)

**S3 Table. Characteristics of included studies.**
(DOCX)

**S4 Table. Main findings of included studies.**
(DOCX)

## Acknowledgments

### Author Note

Dr Robyn Brunton was the sole contributor and responsible for all aspects of this manuscript. The contribution of Dr Denise Corboy, who assisted with the data extraction and quality assessment, is acknowledged.

## Author Contributions

**Conceptualization:** Robyn Brunton.

**Data curation:** Robyn Brunton.

**Formal analysis:** Robyn Brunton.

**Funding acquisition:** Robyn Brunton.

**Investigation:** Robyn Brunton.

**Methodology:** Robyn Brunton.

**Project administration:** Robyn Brunton.

**Writing – original draft:** Robyn Brunton.

**Writing – review & editing:** Robyn Brunton.

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
