## [Decision Letter · Decision Letter 0]

5 Jan 2024

PONE-D-23-17452Childhood Abuse and Perinatal Outcomes for Mother and Child: A Systematic Review of the LiteraturePLOS ONE

Dear Dr. Brunton,

Thank you for submitting your manuscript to PLOS ONE. After careful consideration, we feel that it has merit but does not fully meet PLOS ONE’s publication criteria as it currently stands. Therefore, we invite you to submit a revised version of the manuscript that addresses the points raised during the review process.

**ACADEMIC EDITOR: One of the reviewer considered that the information presented in extensive and recommended separate papers. This is up to the authors to do so. However, this Academic Editor considers the paper can be presented in the current comprehensive fashion if some recommendations are attended indicated by the reviewer:****- Begin each of the separate topical paragraphs with a summary of what they found.****- Any other pertinent considerations to make the manuscript depth easier to follow such as subheadings and figures to provide big picture and roadmap for the reader. ** **However, the manuscript has the adequate rigor and content suitable for publication. These are minor revisions, then. **

We look forward to receiving your revised manuscript.

Kind regards,

Abraham Salinas-Miranda, MD, PhD

Academic Editor

PLOS ONE

Journal Requirements:

Additional Editor Comments (if provided):

Dear authors, minor revisions were requested by one of the reviewer and approved by this academic editor. Please make these minor revisions and submit the revised version as soon as possible.

Reviewers' comments:

Reviewer's Responses to Questions

**Comments to the Author**

1. Is the manuscript technically sound, and do the data support the conclusions?

Reviewer #1: Yes

2. Has the statistical analysis been performed appropriately and rigorously? 

Reviewer #1: N/A

3. Have the authors made all data underlying the findings in their manuscript fully available?

Reviewer #1: Yes

4. Is the manuscript presented in an intelligible fashion and written in standard English?

Reviewer #1: Yes

5. Review Comments to the Author

Reviewer #1: This is very well written and an incredibly thorough review of the literature regarding child maltreatment and perinatal health and outcomes. If anything, there is too much information included in it; as a reader, I felt overwhelmed by the sheer volume of data being conveyed, and it read as if the authors were including absolutely every possible finding from the review, whether it made theoretical sense or was of clinical importance or not (e.g., is there tremendous value in a single study finding that CSA was associated with not attending birthing classes or doing so w/o a partner?).

If the author feels strongly about reporting all of the findings, I wonder if the they could consider dividing the contents into separate papers (e.g., ante/postpartum outcomes or type of outcome).

A second possibility would be for the author to begin each of the separate topical paragraphs with a summary of what they found, and then use SOME of the data as illustrations. I am not convinced that include ALL of the details that are included are necessary; at times these details became cumbersome to follow without re-reading multiple times, especially when the authors were describing multiple studies in the same paragraph. The minutia is really overwhelming. One example is on pg 22, where the author writes "In studies that examined all abuse types, CPA, CSA, and CEA independently predicted PPD; however, only CPA remained significant when controlling for other abuses (30). CEA and CSA were also risk factors for early-onset PPD, and CEA, CPA, and CSA were all risk factors for late-onset. The risk remained for CSA and early-onset PPD and CEA and late-onset PPD after adjusting for sociodemographic factors and other abuse types." Frankly, I got hopelessly lost by the end of the paragraph about what I was supposed to take away.

If the author chooses to keep the paper in the long form that it is, please provide a description at the start of the results section that clearly illustrates how the results will be organized, and make those breaks more evident in the paper itself.

Circling back to methods, why were adolescents excluded? Did you exclude any paper that included both adolescents and adult women? When the author says (pg 6) that "remaining studies were individually assessed due to budget limitations" does this means that of the 90+ articles, 80 were reviewed by one individual? This seems like it could represent a limitation, even if there was good reliability in the 10 articles that were reviewed with the research assistant.

Toward the end of the paper, the authors used "cf." to denote the comparison group; either do this throughout the paper or do not. I found it useful, frankly.

At least twice, the author included information about racial stratification, but it was not clear how this provided useful insight. For example, on Pg 10, they describe a study comparing various types of abused and non-abused women and the outcome of persistent bacterial vaginosis, noting that the association was stronger for Black people compared to non-Black people; is there any theoretical or biological reason why race should impact this association? If not, why report this. If so, say why this matters.

I appreciate the conclusion that all pregnancy-related care should be trauma-informed, and that clinicians need to be aware that childhood experiences with abuse are likely to impact ante- and postnatal health and wellbeing (just as recent trauma is). I also appreciate the critique of current approaches to studying child abuse (e.g., the design of studies needs to be tailored to the population of interest).

Could the author be more clear in the paragraph on pg 32 about the prevalence of different types of abuse being more common in different parts of the world? If different forms of child abuse are not studied as substantially in some parts of the globe (e.g., Africa, South America), then how can there be any certainty that rates are actually higher or lower in these regions? Stigma and discomfort discussing abuse (especially sexual abuse) may be at play, and without asking questions that are specifically intended to measure experiences of abuse in these communities and cultures, prevalence estimates will not be accurate.

I appreciate that in the discussion, the author did summarize the more common findings and occasionally proposed some possible reasons for WHY child abuse may be related to these outcomes. However, some of these pathways are very complex and related to one another, yet the author treats each as if they are siloed (e.g., child abuse and IPV are related; child abuse and depression are related; it seems likely that these factors may influence one another in ways that are difficult for these types of quant studies to get at). This is a limitation of looking at a subject as complex as this is in this way.

In conclusion, this systematic review appears to be very thorough, well done, and well written. But it is easy to lose the forest for the trees with the amount of detail included. Not all of the associations that were identified from the high-quality studies seem to have clinical importance, however, and the amount of detail included often left me unable to discern what WAS important to take away.

6. PLOS authors have the option to publish the peer review history of their article (what does this mean?). If published, this will include your full peer review and any attached files.

Reviewer #1: No

---

## [Editor Report · Decision Letter 1]

3 Apr 2024

Childhood Abuse and Perinatal Outcomes for Mother and Child: A Systematic Review of the Literature

PONE-D-23-17452R1

Dear Dr. Brunton,

We’re pleased to inform you that your manuscript has been judged scientifically suitable for publication and will be formally accepted for publication once it meets all outstanding technical requirements.

Kind regards,

Abraham Salinas-Miranda, MD, PhD

Academic Editor

PLOS ONE

Additional Editor Comments (optional):

The authors have revised the manuscript and responded to all of the reviewers' requests. Notably, the authors re-organized the section of their findings about perinatal outcomes in a clearer fashion by

highlighting key findings (in italics) to show the different outcomes found in the literature. They also provided details of risk of bias, PRISMA, and supplemental tables. The decision is to recommend for publication. This systematic review provides a comprehensive overview of the evidence on the association between child abuse and adverse perinatal outcomes as well as comorbidities.
---

## [Editor Report · Acceptance letter]

8 May 2024

PONE-D-23-17452R1 

PLOS ONE

Dear Dr. Brunton, 

I'm pleased to inform you that your manuscript has been deemed suitable for publication in PLOS ONE. Congratulations! Your manuscript is now being handed over to our production team.

Kind regards, 

on behalf of

Dr. Abraham Salinas-Miranda 

Academic Editor

PLOS ONE